# Determinants of Self-Rated Health Disparities among Independent Community-Dwelling Older Adults: An Age-Stratified Analysis

**DOI:** 10.3390/healthcare11233070

**Published:** 2023-11-30

**Authors:** Yuka Iwata, Ayuka Yokoyama, Nanami Oe, Eriko Ito, Azusa Arimoto, Yuko Tanaka, Etsuko Tadaka

**Affiliations:** 1Department of Community Health Nursing, Graduate School of Medicine, Yokohama City University, Yokohama 236-0004, Japan; 2Department of Nursing Informatics, Graduate School of Nursing Science, St. Luke’s International University, Tokyo 104-0044, Japan; 3Department of Community and Public Health Nursing, Graduate School of Health Sciences, Hokkaido University, Sapporo 060-0812, Japan; 4Department of Community Health Nursing, School of Nursing and Social Services, Health Sciences University of Hokkaido, Ishikari-Tobetsu 061-0293, Japan

**Keywords:** self-rated health, older adults, cross-sectional study

## Abstract

In response to the distinctive healthcare requirements of independent, healthy, community-dwelling older adults in Japan and other developed countries with aging populations, the current study examined the differences in factors associated with self-rated health (SRH) between the following two age groups: young–old (65–74) and old–old (75 and above). Age-stratified analysis was used to provide a comprehensive understanding of the unique health challenges faced by these demographic segments and to inform the development of targeted interventions and health policies to improve their well-being. The results of a cross-sectional study of 846 older adults in Yokohama, Japan, who completed self-administered questionnaires, revealed that high SRH was consistently linked with the low prevalence of concurrent medical issues in both age groups (<75 and ≥75) (β: −0.323, *p* < 0.001 in the <75 group; β: −0.232, *p* < 0.001 in the ≥75 group) and increased subjective well-being (β: 0.357, *p* < 0.001 in the <75 group; β: 0.244, *p* < 0.001 in the ≥75 group). Within the ≥75 age group, higher SRH was associated with more favorable economic status (β: 0.164, *p* < 0.001) and increased engagement in social activities (β: 0.117, *p* = 0.008), even after adjusting for age, sex, and economic status. These findings may inform the development of targeted interventions and policies to enhance the well-being of this growing population in Japan and other developed countries.

## 1. Introduction

Self-rated health (SRH) is a widely recognized and reliable measure of overall health, encompassing various elements such as demographic, cognitive, physical, social, and psychological factors. As the global population ages, it is increasingly important to investigate SRH within the context of older adults who are living longer while coping with multiple chronic conditions. Notably, cardiovascular disease, lung disease, cancer, and stroke have emerged as the predominant causes of mortality in many countries in recent years [1,2,3]. In this context, SRH plays an important role in assessing public health needs at the community level and identifying individuals at risk of declining health at an early stage.

However, despite the growing importance of SRH, research focusing on the old–old age group (aged 75 and above) remains limited in developed countries, even though this age group is rapidly expanding [4]. There is a pressing need to promote research that includes this demographic because understanding the SRH of the old–old is critical for tailoring healthcare and support services to their specific needs.

While prior research has explored changes in SRH over time [5,6], these investigations have predominantly focused on individuals aged 25 to 74 years. However, when we narrow our focus to the older adult population, SRH is intricately connected to the sociocultural context in which individuals are embedded [7]. This raises questions about whether SRH and its associated determinants vary with age in older adults. Additionally, several cross-sectional studies have enrolled a substantial number of older participants grappling with multiple chronic conditions [6,8]. To promote healthy aging, it is important to examine the factors influencing SRH among independent, healthy older adults residing in the community, considering demographic characteristics, health patterns, family dynamics, and sociocultural environments.

To the best of our knowledge, no prior cross-sectional studies have aimed to identify the factors associated with SRH among independent, healthy older adults. Moreover, there is a dearth of research exploring potential age-related disparities in the determinants of SRH within this population. Hence, the current study aimed to examine the factors linked to SRH among independent, healthy, community-dwelling older adults in Japan, with a specific focus on unraveling distinctions in these determinants across different age groups.

In response to the unique healthcare needs of independent, healthy older adults in Japan and other developed countries, the current study aims to clarify disparities in the factors contributing to SRH between the following two distinct age groups: the young–old (aged 65–74) and the old–old (aged 75 and above). This age-stratified analysis is vital for gaining a comprehensive understanding of the distinctive health challenges faced by these demographic segments and can serve as a foundation for the development of targeted interventions and health policies aimed at enhancing their well-being.

The primary objectives of this study were to identify the factors underpinning subjective health, even after accounting for age, sex, and economic status, and to delineate the variations in factors related to SRH across different age groups. This study represents a significant step towards elucidating the multifaceted factors associated with SRH in older adults and the differences in these determinants between age groups.

We hypothesized that the factors contributing to SRH exhibit variations across different age groups, specifically between the young–old (aged 65–74) and the old–old (aged 75 and above). We predicted that age, as a key demographic factor, interacts with other determinants, such as economic status, health patterns, and sociocultural influences, to shape the SRH of independent, healthy community-dwelling older adults. In conducting this study, we aimed to provide insights into the specific factors that impact SRH in these distinct age groups, ultimately contributing to a more tailored approach when addressing individuals’ healthcare needs and enhancing their overall well-being.

## 2. Materials and Methods

### 2.1. Data Source and Study Design

The study population comprised 846 older adults belonging to a community group certified through self-administered questionnaires by the city government of Yokohama, Japan, from 23 October to 29 November 2019. The inclusion criteria for participants were as follows: (1) aged 65 years and over, (2) belonging to a Yokohama-certified community group, (3) independently living without long-term care needs, (4) with the ability and willingness to complete the questionnaire. Community group activities for health promotion in Japan are nationally, institutionally, and culturally driven by local residents, emphasizing population-oriented, preventive, sustainable, and cost-effective approaches over individual-level interventions [9,10]. Therefore, an inclusion criterion (2) was applied. Criterion (3) refers to individuals who did not meet the requirements for certification or the need for long-term care under the Long-Term Care Insurance system in Japan (support need levels are 1 and 2, and care need levels are 1 to 5; a larger number indicates a greater need for care).

The sample size for a multiple regression analysis was calculated using G*Power 3.1.2 on the basis of the moderate effect size of 0.15 (medium), a significance level of 0.05, a power of 80%, and seven predictors (i.e., age, sex, living arrangements, number of concurrent medical issues, economic status, subjective well-being, social activities). Thus, the required sample size was calculated to be 103 for each of the age groups. The final sample size exceeded the pre-study sample size calculation.

### 2.2. Measurements

On the basis of the comprehensive framework of human aging outlined in a previous study [11], which posits that the aging process encompasses biological, psychological, and social dimensions, the present study incorporated a range of demographic characteristics. These characteristics encompass biological factors, including age, sex, and the presence of concurrent medical conditions, as well as psychological factors, such as subjective well-being. Moreover, we also considered social factors, which included economic status, living arrangements, and engagement in social activities. This multi-dimensional approach allowed us to explore the various facets that influence the aging experience more thoroughly.

#### 2.2.1. Age-Stratified Groups

Respondents were divided into two age-stratified groups: those over 75 years old and those under 75 years old. It should be noted that although several researchers in the literature refer to the onset of old age [11,12], there is no universally accepted or consistently applied threshold for the definition of old age. Some studies indicate that the delimitation of old age is marked by the culmination of complex life obligations, the stabilization of life plans, and personal growth. Other studies propose that this transition is characterized by gradual retirement, as individuals in this age group often experience a reduction in income as a consequence of retirement. In the current study, age categories were determined according to the guidelines of the Japanese Geriatrics Society and the Law on Securing Health Care for the Elderly in Japan. According to these guidelines, the 65–74 age group was defined as young–old, and the 75 and older age group was defined as old–old [13].

#### 2.2.2. Dependent Variable

SRH was measured using the visual analog scale (VAS) [14]. The choice of the VAS as a method of self-reporting is grounded in its distinctive capabilities. The VAS exhibits high sensitivity for capturing and quantifying subtle aspects of health. This method is well suited for evaluating subjective and sensory information related to health, enabling participants to reflect on their experiences and perceptions. In the context of health assessment among older adults, the VAS is typically considered an appropriate choice. The reported Cronbach’s alpha was 0.91, and the total scores were significantly correlated with subjective well-being among older adults in Japan [15]. A robust Cronbach’s alpha value is a crucial indicator that the VAS is reliable and consistent in measuring SRH in older adults. This high reliability ensures that the VAS accurately captures and evaluates subtle variations and differences in health status. Overall, the VAS is a reliable and sensitive tool for evaluating SRH among older adults.

#### 2.2.3. Independent Variables

Participants’ characteristics included age, sex, living, concurrent medical issues, and economic status (subjectively sufficient = 1, subjectively insufficient = 0). Economic status was gauged by asking about financial concerns. We anticipated these factors influencing SRH based on prior model findings [16].

Subjective well-being was measured using the five-item World Health Organization Well-Being Index (WHO-5) [17], consisting of the following five dimensions: positive mood, vitality, positive thoughts, clear mind, and interest in daily activities. Higher scores indicate positive mental well-being. Each item, evaluated on a 6-point Likert scale (0 to 25 points), showed a significant correlation with depressive symptoms [17]. SRH serves as a comprehensive health status measure, with our hypothesis emphasizing the need to enhance positive mental well-being for a more thorough assessment [18].

Social activities were measured using the scale of social activities among community-dwelling elderly people (SSAC) [19]. This scale includes six items that measure the frequency of social activities in the community, each using a 3-point Likert scale. Scores range from 6 to 18 points, with higher scores indicating greater participation in social activities. The Cronbach’s alpha value was 0.78, and the total scores were significantly correlated with life satisfaction alongside the frequency with which participants went out [19]. On the basis of findings from previous studies [20,21], we predicted that the decrease in the frequency of going out in cases with lower SSAC scores reflected lower SRH scores. Thus, we used this scale because independent variables influence SRH.

### 2.3. Statistical Analysis

Descriptive analyses were performed to characterize the study population and describe the distribution of SRH and scores stratified by age group. Spearman’s rank correlation coefficient and Chi-squared analyses were used to examine whether there was a difference between SRH and independent variables for each of the age groups. The distributions of the quantitative variables were verified using the Shapiro–Wilk normality test with *p* < 0.05 for all quantitative variables. After identifying significant variables from the Spearman’s rank correlation coefficient and Chi-squared analyses, we conducted a multiple linear regression analysis to the explore key determinants of SRH for each of the age groups. In multiple regression analysis, an adjustment for covariates was conducted for age, sex, and economic status. The multicollinearity of independent variables was considered via the forced entry method. The results were statistically significant with *p* < 0.05. The dependent variable had no missing data compared to the covariate and independent variables. To prevent potential distortions when imputing missing data and recognizing their limited influence on the overall conclusions, the analysis proceeded without data imputation. Analysis utilized IBM SPSS software, version 28.0 (IBM Corp., Armonk, New York, NY, USA).

### 2.4. Ethical Approval

This research was conducted in accordance with the 1964 Declaration of Helsinki (and its amendments) and the ethical guidelines for life sciences and medical research involving human subjects presented by the Ministry of Health, Labour and Welfare of Japan. The Institutional Review Board of the School of Medicine, Yokohama City University (Nos. A201200008 and B220500085).

## 3. Results

### 3.1. Characteristics of the Sample

Table 1 shows the demographic characteristics of study participants. Participants’ mean (SD) SRH score (*n* = 846) was 68.8 (19.2). Participants’ mean (SD) age was 77.7 (5.9) years, and 80.1% of participants were female. 

### 3.2. Self-Rated Health and Levels Stratified by Age Group

Table 2 shows the demographic characteristics in terms of the distribution of age groups. A total of 28.7% of participants (*n* = 243) were younger than 75 years old (<75), and 71.3% (*n* = 603) were 75 years old and older (≥75). Table 2 also shows the results of bivariate analyses. The mean (SD) SRH scores were 69.8 (18.5) in the <75 group and 68.4 (19.5) in the ≥75 group. There was no significant difference in the mean SRH scores between the <75 group and the ≥75 groups (*p* = 0.320). Moreover, the Kolmogorov–Smirnov normality test yielded *p*-values < 0.01 for both groups and did not show a normal distribution in either group, suggesting that the scores were very high for a small number of participants in both the ≥75 group and the <75 groups. On the basis of these results, we decided that it was appropriate to dichotomize the groups in the subsequent multiple-regression analysis.

Table 2 also shows the related factors in the univariate analysis (independent variables) of the SRH. Significant differences in SRH in the <75 group were found to be related to the number of concurrent medical issues (*p* < 0.001), economic status (*p* = 0.002), subjective well-being (*p* < 0.001), and social activities (*p* = 0.004), and significant differences in SRH in the ≥75 group were found to be related to the number of concurrent medical issues (*p* < 0.001), economic status (*p* < 0.001), subjective well-being (*p* < 0.001), and social activities (*p* < 0.001).

### 3.3. Factors Associated with Self-Rated Health Stratified by Age Group—Results of Bivariate Analyses

No multicollinearity was observed among the covariates in any of the models. Table 3 shows the results of the multiple regression analysis while controlling for age, sex, and economic status. In the <75 age group, SRH exhibited significant associations with the number of concurrent medical issues (β: −0.323, *p* < 0.001) and subjective well-being (β: 0.357, *p* < 0.001). In contrast, in the ≥75 age group, SRH displayed significant associations with the number of concurrent medical issues (β: −0.232, *p* < 0.001), economic status (β: 0.164, *p* < 0.001), subjective well-being (β: 0.244, *p* < 0.001), and engagement in social activities (β: 0.117, *p* = 0.008). Interestingly, the results revealed a divergence between the ≥75 age group and the <75 age group in terms of the associations between economic status and social activities with SRH scores. These findings underscore the age-specific variations in the determinants of SRH and provide a foundation for tailored interventions to enhance well-being within these distinct age groups.

## 4. Discussion

This study examined age group differences in factors associated with SRH among independent, healthy, community-dwelling older adults in Japan. To the best of our knowledge, the current study is the first cross-sectional study conducted to determine whether SRH is associated with factors in independent healthy community-dwelling older adults in Japan and whether there is a difference between the following two age groups: 75 years and older, and younger than 75 years.

The results indicated that the participants in this study were representative of independent, healthy, community-dwelling older adults in Japan. First, in terms of demographic characteristics (e.g., age, sex) and the proportion of participants, this study was similar to the reported national statistics for independent, healthy, community-dwelling older adults in Japan [9]. Second, the SRH scores in this study were similar to those obtained when the SRH measure was originally developed and similar to those reported for older adults in Japan [15]. This suggests that the current findings may be generalizable to other older adults, not only in rural areas but also in urban areas, particularly in large cities with district characteristics that are similar to the largest ordinance-designated city (e.g., low levels of co-morbidities), which constituted the survey district for this study, both within Japan and in other countries.

The results from 846 participants in the two age groups demonstrated that higher SRH is significantly associated with a smaller number of concurrent medical issues. These findings are in agreement with the results of a recent cross-sectional study conducted in community-dwelling individuals aged 65–88 years in Iceland, which reported that the likelihood of better SRH decreased with low subjective well-being simulated by depressive symptoms [22]. A study in Brazilian individuals aged 60 years and above also revealed an association between poor SRH and depression in both men and women [23]. The findings of the present study are also consistent with the results reported by Sheridan et al. (2019), who found that multimorbidity combinations, which included high depressive symptoms, were associated with an increased probability of reporting poor SRH [24]. In older adults over 75 years of age, a singular health issue, if it does not limit social activity, may have a minimal impact on an individual’s SRH. Comorbidities were major determinants of SRH, and the prevention of comorbidities greatly reduces the total burden of poor SRH and its consequences, such as poor quality of life and the use of health care services [25].

In the ≥75 groups, a higher SRH was significantly associated with higher economic status. Although an individual’s income does not necessarily decrease when they turn 75 years old, various trends and factors commonly influence SRH through income for older individuals. One relevant factor is an increase in medical costs to maintain health with age. In the 2018 Health and Nutrition Survey, a small percentage of respondents identified monetary costs as barriers to healthy eating habits [26]. Although the results of the current study cannot be compared with those of a national health survey, the present findings suggest that an individual’s economic situation may influence their SRH. Older adults may face economic challenges that limit their ability to maintain their health, potentially affecting their SRH.

Additionally, in the ≥75 groups, higher SRH was significantly associated with social activities. Older adults, especially those over 75 years old, can experience age-related health issues such as chronic conditions, mobility limitations, and a weakened immune system. Engaging in social activities can positively affect physical health in several ways [27]. For example, social activities can encourage older adults to stay active, reducing the risk of obesity and cardiovascular disease [28]. Although some social activities may have very little direct impact on health issues such as obesity and cardiovascular disease, a previous study suggested that social engagement may enhance immune function and reduce the risk of illnesses that could lead to comorbidities [29].

One strength of the current study is that the investigation focused on the concept of older adults who live longer while grappling with medical issues, with an emphasis on examining SRH among independent older adults; the intentional adoption of the age of 75 as the cutoff value to define the parameters for older adults was aimed at clarifying differences in the factors associated with SRH. Distinct patterns emerged when comparing the two age groups. The factors of age (young–old), subjective well-being, and the number of medical issues played pivotal roles in SRH, while the old–old factor exhibited additional associations with economic status and social activities. The current findings suggest that two interventions may be conducive to enhancing SRH among independent older adults. First, an individualized approach aimed at ameliorating mental health issues is supported by the current findings and previous literature [30,31]. This intervention is applicable across different age groups among the elderly population. Second, the current findings suggest the potential usefulness of an environmental approach aimed at fostering social activities and community activation, particularly catering to the old–old demographic aged 75 and above. Such approaches may have implications for informing forthcoming policy initiatives that are geared toward fostering healthy longevity. Furthermore, the strength of the current study lies in its focus on targeting older adults belonging to community groups, which are driven nationally, institutionally, and culturally by local residents. This clarification of participation in community groups is useful for identifying the effective targets of community activities. This may be particularly beneficial for experts who support community activities, potentially informing the development of approaches to strengthen the community and the identification of the types of activities that should be promoted. Additionally, these findings may aid the development of detailed strategies based on the specific characteristics of the target residents.

The SRH data in the current study have significant potential for practical applications by program planners and other stakeholders across various domains. The consideration of how these SRH data can be effectively utilized, leading to the development of targeted interventions, policies, and support programs, includes four key points. The first point is the development of targeted interventions: SRH data serve as an indicator of how older individuals perceive their health. When specific health issues or needs are identified, targeted interventions can be developed. For instance, programs or services addressing particular physical or mental health challenges can be tailored on the basis of the identified SRH trends. The second point is policy formulation and prioritization: SRH data provide comprehensive insights into the health status of older individuals. This information can be used to formulate policies and prioritize initiatives at the community or facility level. For example, policies could be designed to focus on groups maintaining a high SRH, or conversely, specific strategies can be devised for those who have a lower SRH. The third point is the deployment of preventive programs: SRH data offer clues for the early detection of health issues. This can inform the deployment of preventive programs and health promotion activities. Screening or educational programs can be implemented for groups with specifically identified health risks. The fourth point is building community support: SRH data aid in the understanding of the overall health status of the community. This insight can guide the development of community-wide support programs and resources. Initiatives addressing specific health-related needs of the community can be promoted.

The current study involved several limitations that should be considered. First, the current study design meant that we were not able to establish causal relationships between the SRH scores, demographics, and psychosocial factors. Therefore, further longitudinal and intervention studies are needed to clarify this issue. The second limitation is related to the study population. The target population was located in the largest ordinance-designated city in Japan, and the results should be interpreted with caution in their application to the Japanese population as a whole, particularly those living in rural areas. Therefore, replication studies with international population-based samples would be useful to confirm the current findings.

## 5. Conclusions

This study investigated the factors associated with SRH among independent, healthy, community-dwelling older adults in Japan and assessed the age-specific distinctions in these determinants. Importantly, the present findings reveal two crucial insights: (1) in both age groups, a higher SRH was consistently and significantly linked to a reduced number of concurrent medical issues and an elevated sense of subjective well-being; (2) within the ≥75 age group, a higher SRH was associated with higher economic status and greater engagement in social activities. These findings significantly expand our current understanding of the subject, shedding light on the intricate interplay of these factors in shaping SRH among older adults. Furthermore, these findings contribute to the development of targeted interventions and policies aimed at enhancing the well-being of this growing population in Japan and other developed countries.

## Figures and Tables

**Table 1 healthcare-11-03070-t001:** The demographic characteristics of the study participants.

		*n* = 846
Variable	Mean ± SD or *n* (%)	
Self-rated health ^a^	68.8 ± 19.2	
Age	77.7 ± 5.9	
Sex		
Female	678	(80.1)
Male	154	(18.2)
Missing	14	(1.7)
Living arrangements		
Alone	234	(27.7)
With spouse	306	(36.2)
With children	111	(13.1)
With spouse and children	39	(4.6)
With children and grandchildren	133	(15.7)
Others	20	(2.4)
Missing	3	(0.4)
Number of medical issues	1.3 ± 1.0	
0	166	(19.6)
1	369	(43.6)
2	204	(24.1)
3	83	(9.8)
4	18	(2.1)
5	3	(0.4)
6	1	(0.1)
7	0	(0.0)
Missing	2	(0.2)
Economic status		
Sufficient	732	(86.5)
Insufficient	109	(12.9)
Missing	5	(0.6)
Subjective well-being scores ^b^	16.4 ± 5.1	
Social activities scores ^c^	12.3 ± 2.9	

^a^ Self-rated health: the visual analog scale (VAS) [14,15] ^b^ Subjective well-being: the five-item World Health Organization Well-Being Index (WHO-5) [17] ^c^ Social activities: the scale of social activities among community-dwelling elderly people (SSAC) [19].

**Table 2 healthcare-11-03070-t002:** Distribution of demographic characteristics in the two age groups.

						*n* = 846
Variable	<75 years *n* = 243			≥75 years *n* = 603		
	Mean ± SD or *n* (%)		*p*-value	Mean ± SD or *n* (%)		*p*-value
Self-rated health	69.8 ± 18.5		-	68.4 ± 19.5		-
Age	70.5 ± 2.5		n.s.	80.3 ± 3.9		n.s.
Sex			n.s.			n.s.
Female	200	(82.6)		478	(81.0)	
Male	42	(17.4)		112	(19.0)	
Living arrangements			n.s.			n.s.
Alone	43	(17.7)		191	(31.8)	
With spouse	109	(44.9)		197	(32.8)	
With children	28	(11.5)		105	(17.5)	
With spouse and children	45	(18.5)		66	(11.0)	
With children and grandchildren	10	(4.1)		29	(4.8)	
Others	8	(3.3)		12	(2.0)	
Number of medical issues	1.2 ± 1.0		<0.001 **	1.4 ± 1.0		<0.001 **
0	65	(26.9)		101	(16.8)	
1	107	(44.2)		262	(43.5)	
2	49	(20.2)		155	(25.7)	
3	17	(7.0)		66	(11.0)	
4	4	(1.7)		14	(2.3)	
5	0	(0.0)		3	(0.5)	
6	0	(0.0)		0	(0.0)	
7	0	(0.0)		1	(0.2)	
Economic status			0.002 **			<0.001 **
Sufficient	201	(83.8)		531	(88.4)	
Insufficient	39	(16.3)		70	(11.6)	
Subjective well-being scores	16.5 ± 4.7		<0.001 **	16.5 ± 5.2		<0.001 **
Social activities scores	12.6 ± 3.0		0.004 **	12.3 ± 2.8		<0.001 **

n.s.: not significant, **: *p* < 0.01. Subjective well-being: scores on the five-item World Health Organization Well-Being Index (WHO-5) [17]. Social activities: scores on the scale of social activities among community-dwelling elderly people (SSAC) [19]. Missing data were excluded.

**Table 3 healthcare-11-03070-t003:** Factors associated with self-rated health stratified by age group.

						*n* = 846
Variable	<75 years *n* = 243			≥75 years *n* = 603		
	Partial regression coefficient	β	*p*-value	Partial regression coefficient	β	*p*-value
Sex	−3.460	−0.070	0.209	0.235	0.005	0.906
Economic status	5.225	0.101	0.080	9.852	0.164	<0.001 **
Number of medical issues	−6.127	−0.323	<0.001 **	−4.212	−0.232	<0.001 **
Subjective well-being	1.398	0.357	<0.001 **	0.889	0.244	<0.001 **
Social activities	0.234	0.038	0.521	0.782	0.117	0.008 **
adjusted R^2^			0.330			0.202

*β*: Standard partial regression coefficient; **: *p* < 0.01. Sex: male = 1, female = 0, Economic status: subjectively sufficient = 1, subjectively insufficient = 0. Subjective well-being: scores on the five-item World Health Organization Well-Being Index (WHO-5) [17]. Social activities: scores on the scale of social activities among community-dwelling elderly people (SSAC) [19]. An adjustment for covariates was conducted for age, sex and economic status. Missing data were excluded.

## Data Availability

The data that support the findings of this study are available from Yokohama City Government and Yokohama City University, but restrictions apply to the availability of these data under the Japan Personal Information Protection Law, which were used under license for the current study, and so are not publicly available. The data are, however, available from the first/corresponding authors upon reasonable request and with the permission of the Yokohama City Local Government and Yokohama City University.

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
