# Peer review of "Determinants of Self-Rated Health Disparities among Independent Community-Dwelling Older Adults: An Age-Stratified Analysis"

_healthcare, 2023, doi:10.3390/healthcare11233070_

Round 1

Reviewer 1 Report

Comments and Suggestions for Authors Dear Author,
I find your paper very interesting. There are some comments and suggestions, so please correct the manuscript.
1. In Abstract and Methods: Actually, there are not 1,900 participants in the study. You included 846 older adults in the research. Please correct this. 2. Line 105: You wrote „seven predictors“ - define these 3. Discussion: line 254: you point out that " current findings may be generalizable to the older adults in Japan, and other countries“, but in line 305. you point out in the limitation of the study that "the target population was located in the largest city ... and that the results should be interpreted with caution in their application…" This is the opposite of the first statement. Please make this clearer. 5. In the discussion section, it is not necessary to repeat the values from resulta /line 257, 269 and 279. Please remove them

Author Response

The reviewer comments were highly insightful and enabled us to greatly improve the quality of our manuscript. We include our point-by-point responses to each of the comments of the reviewer as well as your own comments. Please see the attachment.

Reviewer 2 Report

Comments and Suggestions for Authors

This study examines how age (below vs above age 75) and a host of health and social factors are related to Self Rated Health among a sample of Japanese community organization members in a large city. The study found no main effect of age ofn SRH but it did find interactions between age and several of the other independent variable. The study is well-presented and could benefit from a more developed theoretical framework. There are a number of issues that should be addressed in a revision.

1) Please say more about the importance of membership in a community organization to the characteristics of the sample. How should such membership influence interpretation of findings? The authors also claim that the same is representative of the Japanese elder population. It would be helpful to provide more of the evidence for this statement.

2) The dependent variable needs to be described more: there is a reference to a Chronbach's alpha but no description of the component measures. The bivariate relationships between SRH and subjective well-being also need further discussion. Do these scales really intend to measure different constructs?

3) The analysis as presented in Table 3 should be much clearer.....I think you tested the main effect of age on SRH controlling for the other independent variables. And then you examined how the other independent variables (and age used as a continuous variable) within the two age groups. This analysis does not specifically test the interaction of age with the other independent variables on SRH. Since you hypothesized interactive effects, it is appropriate and needed observed differences in determinants of SRH by age group may not be significant.

4) I do not feel the discussion provides an adequate assessment of the practical significance of the SRH (or for that matter, subjective well-being). Knowing these scores provides some indication of individuals and groups who are experiencing less joyful and dignified aging, yet how program planners and others would utilizes these data might be made clear. 

Comments on the Quality of English Language

The paper is well written and free of grammatical and usage errors.

Author Response

(The authors gave the same response as above.)

Reviewer 3 Report

Comments and Suggestions for Authors

This paper sets out to describe ‘the differences in factors associated with Self-Rated Health (SRH) between two 18 age groups: young-old (65–74) and old-old (75 and above)’. Using a self-administered questionnaire to members of a ‘community group’, the authors conclude that there appear to be two key factors associated with better well-being: favourable economic circumstances and engagement in social activities.

Overall the paper is very well constructed, argued and written and equally important of value for policy makers when considering what is likely to be a substantial challenge into the future. There are a few minor points that I would suggest the authors should consider prior to acceptance.

First, the authors describe the sample as being derived from ‘belonging to a community group’. This implies that there is already a level of engagement for the sample. The paper would benefit from clarification of what the term ‘community group’ means and equally importantly, what the groups’ functions are. This is important as the findings may have considerable relevance for other countries besides Japan.

Second, the sample description needs to be placed in the wider population context. The paper would be strengthened by this process as the levels of co-morbidities appear to be relatively low when compared to other developed countries for the age groups reported. Some aspects of this are touched upon in what the authors have labelled as ‘limitations’. 

Third, the paper would benefit from a brief description of ‘comorbidities’ and ‘social activities’. For the former, a single condition could cover a range of conditions which might impact of the individual’s activities and another, have very little impact. For the latter, ‘social activities’ also covers a range, some of which may have very little impact on obesity of cardio-vascular disease directly. 

Overall however, the authors are to be congratulated on their work and the paper in what is an important area for a considerable number of countries. 

Author Response

(The authors gave the same response as above.)
